# The Role of the RNA-RNA Interactome in the Hepatitis C Virus Life Cycle

**DOI:** 10.3390/ijms21041479

**Published:** 2020-02-21

**Authors:** Cristina Romero-López, Alfredo Berzal-Herranz

**Affiliations:** Instituto de Parasitología y Biomedicina López-Neyra (IPBLN-CSIC), Av. Conocimiento 17, Armilla, 18016 Granada, Spain

**Keywords:** HCV, interactome, functional RNA domains, long-distant RNA-RNA interactions

## Abstract

RNA virus genomes are multifunctional entities endowed with conserved structural elements that control translation, replication and encapsidation, among other processes. The preservation of these structural RNA elements constraints the genomic sequence variability. The hepatitis C virus (HCV) genome is a positive, single-stranded RNA molecule with numerous conserved structural elements that manage different steps during the infection cycle. Their function is ensured by the association of protein factors, but also by the establishment of complex, active, long-range RNA-RNA interaction networks-the so-called HCV RNA interactome. This review describes the RNA genome functions mediated via RNA-RNA contacts, and revisits some canonical ideas regarding the role of functional high-order structures during the HCV infective cycle. By outlining the roles of long-range RNA-RNA interactions from translation to virion budding, and the functional domains involved, this work provides an overview of the HCV genome as a dynamic device that manages the course of viral infection.

## 1. Introduction

The function of RNA molecules is dependent on their sequence, but also on their structure. The great flexibility and dynamism of RNA folding underlies the versatile features of RNA elements. RNA structure is maintained by the widespread establishment of hydrogen bonds between different nucleotides, yielding a set of loops and stems that define the overall folding of the molecule. These help generate complex conformations, such as helices duplexes, triple-stranded structures and loop-loop connections. Ultimately, they allow for intricate networks of RNA-RNA interactions, the so-called interactome, which generates dynamic, high-order RNA structures [1]. These networks of contacts include both intra- and intermolecular connections, adding a new level of complexity to the architecture of the RNA genome. The interactions that occur generate different RNA structures that perform tasks added to the storage of information [2,3]. Understanding RNA folding has, therefore, become an area of major interest, in which further knowledge is sought on the role of RNA in transcription elongation, splicing, translation and the synthesis of different protein isoforms, etc. [4].

Viral RNA genomes are compact entities with the capacity to perform different functions during the infective cycle, as well as carrying protein-coding information. Such genomic plasticity confers significant adaptive capacity, allowing new hosts and molecular contexts to be explored, thus, promoting the emergence of new diseases. Given their high mutation rates and population sizes, the evolutionary capacity of RNA viruses favors the appearance of variants that are resistant to the host immune system and to treatment. This is achieved by the viral genome, increasing its sequence variation without disturbing functions essential to the execution of the viral cycle [5]. RNA viruses have evolved to acquire a supra-coding information system defined by discrete, complexly folded and highly conserved RNA structural/functional domains that overlap with the protein-coding sequence. These domains interact with proteins and other macromolecules, and establish long-distance interactions with other genomic RNA elements to control the synthesis of viral proteins, the replication of the viral genome, and its encapsidation [2,3].

Hepatitis C virus (HCV) has been studied as a model member of the genus *Hepacivirus*, family Flaviviridae. HCV was discovered three decades ago as the major causal agent of post-transfusion non-A, non-B hepatitis [6]. From this moment, the rapid emergence of serological and nucleic acid-based diagnostic tools facilitated the screening of blood samples and the identification of infected patients. Treatments with direct acting antivirals (DAAs) against viral proteins, either alone or in combination with pegylated interferon-α and ribavirin, now allow the infection to be controlled, significantly improving patient prognosis (for a review, see [7]). Further, the recent development of pangenotypic DAAs is particularly interesting since it provides direct treatment without previous genotype testing. Despite these advances in therapeutics, many of the molecular mechanisms underlying the control of the infective cycle remain unknown. For many years, the lack of robust cell culture systems, along with difficulties in recognizing the actual phase of the life cycle in operation, represented major obstacles to gaining a complete overview of the infection process. Nowadays, workable replication culture and infectious systems provide a complete picture [8,9]. Along with these advances, the development of the “omics” field in molecular biology has led to powerful high-throughput methodologies that return huge amounts of data from just a single experiment. These advances have filled the gaps in our knowledge of some virus cycle control systems, and have revealed the importance of long-range RNA-RNA interactions in the intracellular phase of infection. Further, they have shown that other RNA viruses, such as flaviviruses [10], or the distant retroviruses [11], share important molecular features with HCV. Hence, the study of the molecular mechanisms mediated by the HCV RNA genome has contributed greatly to our understanding of other viral infections.

This review summarizes the major achievements made in our understanding of the HCV genomic RNA interactome. In particular, recent findings in the field of HCV genomic structure are traced to define the roles of the alternative conformations of the genomic RNA in the progression of the intracellular infection. An overview of the participation of the HCV RNA genome in all stages of the virus life cycle is provided, and the ubiquity of long-range RNA-RNA interactions in other viral models discussed.

## 2. The HCV RNA Genome Is a Compact and Resourceful Entity

Like most RNA viruses, HCV is highly variable from a genetic point of view, with eight different genotypes showing more than 30% nucleotide sequence divergence among them [12,13]. Genotype 1, with the subtypes 1a and 1b, is the most prevalent variant and the most resistant to treatment and host defenses, and causes about 40% of infections. These subtypes have been the most studied and used as molecular models.

The HCV RNA genome is an ~9.6 kb-long, single-stranded, positive RNA molecule, that encodes a single open reading frame (ORF) flanked by highly conserved untranslated regions (UTR) [6,14,15]. The translation of the ORF generates the structural proteins present in the virions, including the capsid protein (C), the envelope proteins E1 and E2, and the p7 protein, as well as non-structural products involved in polyprotein processing (NS2, NS3, NS4A) and replication (NS4B, NS5A and NS5B) (Figure 1A). The synthesis of an additional viral gene product has been described, also initiating from the AUG codon (position 342), but with ribosomes shifting at the eleventh codon towards an alternative reading frame to yield the F protein (Figure 1A) [16,17,18,19].

The viral RNA genome is clearly a versatile device. Throughout the genome, structurally preserved RNA domains form groups of functional and regulatory active regions that play important roles in the execution of the infective cycle [20,21,22]. The search for these structural and functional domains has spanned two decades. By using high-throughput techniques, classical biochemical methodologies, viral genetics techniques, comparative sequence analyses and bioinformatics strategies, different groups have provided important information on a large set of structural domains (Figure 1B) [20,21,22,23].

Some of the most conserved domains, both in sequence and structure, map within the UTRs, the core coding sequence, and the 3′ terminus of the ORF (the hypervariable –HV- stem-loop in the 3′UTR is the most variable element) (Figure 1B) [20,22,24,25]. The high sequence and structure conservation rate suggest that these regions have been preserved since the early HCV ancestors of current HCV lineages [20]. Additionally, many well-defined structures have been identified in the central part of the genome that does not involve sequence conservation (Figure 1B) [20]. Interestingly, these conserved elements do not emerge randomly in the ORF; indeed, they are located in precise positions and work to constrain genomic variation [21,22,23,25,26]. It has been suggested that the preservation of these elements in well-defined regions of the HCV genome facilitates virus persistence by avoiding the recognition and degradation mediated by RNase L, an innate intracellular antiviral defense mechanism [27]. RNase L is a potent endoribonuclease that cleaves ssRNA at UA and UU dinucleotides; it is induced by IFN. It is noteworthy that HCV RNA genomes corresponding to the most resistant genotypes have a lower proportion of these dinucleotides compared to variants that are more susceptible to treatment [28]. As well as protecting against RNase L-mediated degradation, some stem-loops identified in the central part of the genome have a regulatory function, which they execute by swapping between different conformational states [20,22,26]. Therefore, the HCV genome evolves under different pressures to maintain its compact folding based on short stem loops that can interact with one another to increase viral fitness.

The genomic sequence not involved in the formation of conserved structures shows significant variation, a consequence of the high mutation rate associated with the action of the viral RNA polymerase (nucleotide changes per replication round occur at a rate of ~1 in 10,000 [29]). While this could have potentially drastic consequences for virus survival, it has been demonstrated that, on average, only around 10% of the genomic sequence is subject to positive selection. Many studies have reported extensive positive selection to operate only on the E1, E2 and NS5A genes [30,31]. The increase in the mutation rate at these locations correlates perfectly with the important role of their corresponding protein products in the viral response to the host immune system and to therapeutic agents, thus, favoring the emergence of viral variants and the persistence of the infection [32,33]. Further, high mutation rates agree perfectly with the ability of HCV to diversify in infected patients [33].

## 3. The hepacivirus Life Cycle

The HCV life cycle is only partially understood, due to the absence of suitably robust cell culture and in vivo infection models. In addition, the complex network of surface receptors involved in efficient viral internalization has delayed the complete understanding of many different molecular aspects of the HCV intracellular cycle. Figure 2 shows the currently accepted model.

The genome in HCV virions is covered by a lipid membrane in which the viral glycoproteins E1 and E2 are embedded. HCV particles associated with neutral lipids and apolipoproteins in the bloodstream, which helps in cell selection and attachment during virus entry [34]. HCV internalization is a complex, multistep event regulated by numerous interactions involving the glycoproteins E1 and E2, and by different cellular receptors [35]. Following attachment to liver cells, the viral particles are internalized via the clathrin-mediated endocytic pathway.

Upon entry, the disruption of the capsid in the endosome allows for the release of the RNA genome into the cytoplasm. Here, the viral RNA is translated to yield a single polypeptide that is co- and post-translationally processed by cellular and viral proteases to generate structural and non-structural (NS) proteins [36] (Figure 2). Most of the NS proteins associated with the endoplasmic reticulum membranes to constitute the membranous web required for the formation of the replication complex [37]. The positive RNA genome serves as a template for the synthesis of the negative strand via the action of RNA-dependent RNA polymerase, which is encoded by the NS5B gene. Interestingly, the negative strand remains base-paired with its template and can be detected in the cytoplasm of infected cells in very small amounts. The resulting dsRNA, the so-called replicative form, is then amplified in a semiconservative and asymmetric manner to generate five to ten-fold molar excess of the positive stranded RNA progeny [38]. This progeny is devoted to new rounds of translation-replication, or packaged into new viral particles that are released into the extracellular medium (for a review, see [7]).

## 4. Translation in HCV Requires Functional Genomic Domains

Since the virion bears only a single positive strand RNA genome, translation is the first step to be accomplished during the intracellular phase. Viral protein synthesis depends on the cell machinery, but it is initiated by a cap-independent, non-canonical pathway that involves an internal ribosome entry site (IRES) (Figure 2) [39]. The HCV IRES is a structurally unique element that uses dynamic RNA domains as scaffolds for the recruitment of the translation machinery, minimizing the requirements of protein factors. Thus, the IRES helps to bypass the canonical cap-mediated screening control to reduce the cellular response to infection.

The most widely accepted model of HCV IRES-mediated translation initiation can be divided into three well-defined steps [40,41]. In the first, the IRES directly binds the 40S particle via a high-affinity mechanism to yield the binary complex [42,43]. Secondly, this binary complex recruits the eIF3 and the Met-tRNAiMet-eIF2-GTP ternary complex to generate the 48S pre-initiation complex [44,45,46,47]. It should be noted that eIF3-IRES binding is a decoy tool to displace the eIF3 from the canonical 43S pre-initiation translation complex, thus, releasing the IRES binding site in the 40S ribosomal subunit [48]. Therefore, it is not essential for the initiation of viral translation per se. Finally, initiation is completed by the joining of the 60S subunit promoted by GTP hydrolysis, which depends on eIF5B, and the dissociation of the above factors from the pre-initiation translation complex [49]. Importantly, and in a manner similar to the canonical initiation and elongation steps, Met-tRNA must base pair with the translation start codon in the P site [49].

HCV can adapt to different cellular stress situations to overcome eIF2 inactivation or increases in Mg^2+^ concentration [50,51,52,53]. This is achieved by reducing the need for translation initiation factors, while the assembly of the 40S and the 60S subunits is still preserved [54]. Alternatively, the HCV IRES binds directly to pre-assembled translation pre-initiation complexes and induces a conformational remodeling to accommodate the system to its own needs [53]. Thus, HCV IRES is a heterogeneous, conformationally complex element, which might explain the virus’ adaptability to the cellular conditions it encounters.

The IRES maps within the 5′UTR and also spans a short stretch of the coding sequence (Figure 3) [39,43,55,56]. Although the secondary structure of the IRES has been extensively studied for two decades, high-resolution structural analyses have recently defined its three-dimensional folding in the absence and presence of the 40S subunit, providing new insights into how RNA folding may reduce the need for translation initiation factors [57,58,59,60,61,62]; the next section explores this further.

### 4.1. Functional RNA Domains Required for Binary Complex Formation

The first step of viral translation requires the direct recognition of the 40S subunit. Under physiological magnesium conditions (~0.5 mM) [63,64], the HCV IRES region binds to the 40S ribosomal subunit [64]. This binding is directed by a stable and specific tertiary structure defined by two major and coaxially stacked extended domains - domains II and III - which are organized around a compact double-pseudoknot element (PK1 and PK2; Figure 3) [57,60,65].

Using a combination of structure probing in solution, cryo-electron microscopy, molecular dynamics and bioinformatics modelling, different authors have observed that the formation of the binary complex depends on direct interaction between subdomain IIId, within the IRES, and 18S rRNA [48,58,66,67]. Subdomain IIId maps within the highly branched domain III of the IRES and is phylogenetically conserved from a sequence and structural point of view [25,68]. It is a 27 nt-long hairpin, with the stem interrupted by an asymmetric internal E-loop, and is capped by a hexanucleotide loop (Figure 3). The E-loop structure is conserved among different isolates, and its stability depends on the formation of AA/AG pairs and a reverse Hoogsteen interaction [69]. In nuclear magnetic resonance analyses, the apical loop appears as a disordered region in which the 5′ side is stacked, while the 3′ side is exposed to the solvent to generate a backbone reversion known as a U-turn [69]. This conformation creates an exceptional structural and sequence environment essential for the formation of the IRES-40S complex [48,64,66,67,69]. In fact, binding to the 40S subunit depends on RNA-RNA interactions involving the phylogenetically conserved GGG triplet within the apical loop of subdomain IIId, and the complementary CCC triplet of helix 26 in the 18S rRNA [48,58,66,67]. The essential role of the apical loop of subdomain IIId in virus survival, therefore, relies on this interaction. This idea fits perfectly with the results of phylogenetic and functional studies that show that mutations in the apical loop drastically reduce IRES activity in vitro and in cell culture [41,68,70,71,72]. In addition, the apical loop of subdomain IIId might be related to differential responses to interferon treatment, although additional factors might contribute to the outcome of therapy [72]. The molecular mechanisms underlying the activity of IIId may, therefore, be based on the combination of both protein and RNA recruitment, suggesting subdomain IIId to be an important element in the HCV interactome. Together, these results reinforce the importance of sequence and structure preservation in assessing the biological role of subdomain IIId, and make it a tempting therapeutic target [73].

Domain II of the IRES also plays an important role as a conformational manager in the constitution of the IRES-40S binary complex [74]. This domain contacts with the 40S subunit over the decoding groove, promoting changes in its structure in a manner similar to those induced by eIF1 and eIF1A in the canonical eukaryotic translation model [61,62,74,75]. Domain II folds into a distorted stem-loop with an L-shape, a consequence of the presence of the internal E-loop (Figure 3) [76,77]. This E-loop creates a bend that separates the basal stem IIa and the apical stem-loop IIb, which directly contacts with ribosomal proteins S14 and S16 [78] at the interface between the 40S and 60S particles in the mRNA binding cleft. When bound to the 40S subunit, a complex four-way junction defined by the double-pseudoknot organizes domains II and III in a coaxial manner [46]. In this conformation, domain III binds to the solvent side of the 40S particle, while domain II reaches the interface cleft and the E-site of the ribosome, leading to a rotation of the ribosomal head and the consequent opening of the mRNA entry channel [57,74,79]. Such a structural organization is also managed by the coordinated action of long-range interactions within domains of the IRES. These interactions mainly involve domains II and IV (a short stem-loop containing the AUG codon) (Figure 3 and Figure 4) [80,81]. Thus, subdomain IIb favors the unwinding of domain IV to accommodate it within the 40S decoding groove, efficiently initiating viral translation [46].

### 4.2. Structural Basis for the Recognition of eIFs by the HCV IRES

The constitution of the 48S complex requires the recruitment of eIF2 and eIF3 to the IRES-40S complex [82].

eIF3 specifically recognizes the apical portion of domain III, involving the subdomains IIIa, IIIb and IIIc (Figure 2 and Figure 3) [83]. These subdomains are organized into a four-way junction that operates as a platform for factor association [47,69,83,84]. Interestingly, this region of the IRES accumulates a certain degree of sequence variation, rendering viral variants with different translational efficiencies [85]. This finding supports the evidence that the association of eIF3 is not essential for viral protein synthesis [48]. In addition, the bond between the IRES and eIF3 is weaker than that between the IRES and the 40S subunit [40], favoring the chances of domain III positioning itself correctly in the pre-initiation complex [48]. The association or release of eIF3 by the IRES can be conditioned, at least in part, by the local flexibility of the GC-rich stem within subdomain IIIb (Figure 3) [83,86]. This subdomain is composed of a long stem, which is interrupted by a conserved CC mismatch and the variable internal loop, and closed by a large and heterogeneous apical loop. As revealed by mutagenesis studies, the preservation of the sequence within the apical loop is not essential for IRES activity, but it could influence different mechanisms of translation control [87,88], reinforcing the role of the IRES three-dimensional structure as the main control element during translation initiation.

### 4.3. Dynamic Conformational Tuning of the Translationally Active 80S Complex Mediated by IRES Domains

Different high resolution microscopy studies have provided a dynamic view of the IRES-80S complex [61,62]. The recruitment of the 60S particle requires the conformational rearrangement of the 40S subunit, not only in terms of the ribosomal proteins, but also of the 18S rRNA [58,61]. These rearrangements are reverted when tRNAi occupies the P site. The reversion is coincident with a structural reorientation of domain II [62]. Thus, domain II moves out of the E site to allow for the incoming deacylated tRNA from the P site. As a consequence of these reorientation events, eIF5-dependent eIF2-GTP hydrolysis is activated. The translation pre-initiation factors are then released from the ribonucleoprotein complex, and the productive translation initiation complex is constituted.

Translation initiation in HCV is, thus, a dynamic and highly controlled process dependent on structural units that operate in a coordinated fashion. Such dynamism may also favor the existence of different pathways for overcoming adverse environmental conditions.

### 4.4. Translational Enhancement Mediated by cis-acting RNA Elements and Long Range RNA-RNA Contacts

High-order RNA structures affect translation initiation, elongation and termination. In particular, mRNA circularization increases translational efficiency by encouraging ribosome recycling and providing protection against the action of exonucleases [89]. In viral RNA genomes, the acquisition of a closed-loop conformation and the establishment of long-range interactions between different functional genomic domains work as efficient and flexible managers of multiple stages of the intracellular virus cycle.

HCV RNA has acquired a sophisticated translational control system. The molecular mechanism involves three differently evolved functional regions (Figure 2 and Figure 4A): (i) Subdomain IIId and the core double-pseudoknot in the IRES (the role of the latter in viral protein synthesis has already been detailed); (ii) domain 5BSL3.2 at the 3′ end of the ORF; and (iii) the 3′X-tail at the very 3′ end of the HCV RNA genome.

The acquisition of a circular isoform by the HCV genome during translation initiation is enhanced by the likely oligomerization of the polypyrimidine tract-binding (PTB) protein, which binds to both the IRES and the 3′X tail [89,90,91]. Furthermore, proteins might be involved in this process [92,93]. The 3′X directly recruits different ribosomal components and the eIF3 protein (Figure 2) [93,94], supporting a model in which translational enhancement occurs in *cis* via the transfer of the translation machinery from the 3′ end of the genome to the IRES, especially during the termination stage of each translation round.

The 3′X tail - a 98 nt-long region showing strong sequence and structure conservation (Figure 5) - was identified almost simultaneously by different groups [95,96]. The fact that only a single substitutions in the 3′ terminal region have been detected suggests it has an important function in the infective cycle [96]. From a structural point of view, the 3′X element shows dynamic behavior by adopting two major and mutually exclusive conformations of similar thermodynamic stability (Figure 5). Both isoforms preserve the 3′SL1 stem loop at the very 3′ end of the viral genome, while the 55 nt-long upstream region changes from two stem-loops (3′SL3 and 3′SL2) to a single, extended stem-loop, i.e., 3′SL2′. The latter exposes a palindromic nucleotide sequence (dimer linkage sequence, DLS) in the apical-loop (Figure 5) [90,97,98,99]. It has been suggested that the switch between conformations may be related to differential ligand (protein and RNA) affinity. In fact, the binding of the PTB protein, ribosomal components and eIF3 seems to occur largely at 3′SL2, suggesting that the three stem-loop isoform may operate as a regulatory element at the translational level [90,93].

A long-range RNA-RNA interaction that seems to be required for efficient HCV translation has also been reported (Figure 2) [100]. This interaction involves the k-motif in 3′SL2 in the 3′X tail, and the complementary nucleotide sequence k’ in the upstream 5BSL3.2 domain [99,101,102,103,104], yielding a dynamic pseudoknot structure (Figure 4A and Figure 5) [104]. Domain 5BSL3.2 is located at the 3′ end of the ORF within the NS5B coding sequence; it is a 48 nt-long imperfect stem-loop with a 12 nt-long apical loop (Figure 5) and an 8 nt-long bulge interrupting the duplex [102,103,105]. Both unpaired regions within this domain are phylogenetically conserved across different genotypes [102], suggesting their participation in the establishment of interactions with other regions of the viral genome [106]. The acquisition of a closed-loop conformation defined by the specific contact 5BSL3.2-3′SL2 seems to be essential for the enhancement of translation, most likely by inducing the opening of 3′SL2 [100,104]. This might favor the recruitment of factors required by the cell translational machinery. Abolishing the 5BSL3.2-3′SL2 contact reduces translation [100]. This regulatory mechanism gains further relevance in the light of results showing that the contact 5BSL3.2-3′X occurs in the absence of RNA chaperone proteins and even in the two stem-loop conformation of the 3′X, in which the k motif is partially occluded in the stem of 3′SL2 (Figure 5) [107]. This points to domain 5BSL3.2 acting as a structural cofactor that promotes the conformational rearrangement of the 3′ end of the viral genome. The described regulatory system, thus, has a riboswitch-like mode of action which, depending on the presence of specific ligands or external stimuli, allows for two mutually exclusive metastable structural states. This device ensures that adequate viral protein levels are in place before the replication and encapsidation steps begin.

## 5. From Translation to Replication. The Role of HCV RNA Circularization

HCV replicates in the cytoplasm once viral protein levels are adequate. Actively translating viral RNA molecules must be transferred to the replication complexes, which contain non-structural viral proteins embedded in the endoplasmic reticulum. This supramolecular complex generates a membranous web that provides a suitable microenvironment for replication [108,109]. In the replication complexes, positive ssRNA acts as a template for the synthesis of the full length negative ssRNA intermediates required to generate the positive ssRNA genomes for use in new rounds of translation, replication and virion packaging. Since RNA replication has to initiate from the 3′- end of the RNA template, this region accumulates numerous functional domains that participate not only in translation (see above), but also in replication control.

The molecular mechanisms underlying the switch from translation to replication in HCV are largely still only hypotheses. Plausible proposals involve the participation of genomic domains acting in *cis*, both as binding platforms for cellular and viral proteins and as organizers of long-distance RNA-RNA contacts (Figure 2 and Figure 4B).

The recruitment of viral NS5B polymerase by the 5BSL3.2 domain has been reported, [110], presumably to form the replication complex in combination with other non-structural HCV proteins and cellular factors, such as ribosome components [111]. The demonstrated ability of NS5B to bind ribosomes reflects the mechanism used by Qβ virus to build its own replication system [111]. Interestingly, the ribosome bound-NS5B forms are highly active in RNA replication [112]. From a functional and physical point of view, both NS5B and ribosomes are associated via the highly conserved domains 5BSL3.1 and 5BSL3.3 that flank the 5BSL3.2 domain (Figure 5) [113]. Domains 5BSL3.1, 5BSL3.2, and 5BSL3.3 define an element located at the 3′ end of the ORF with regulatory roles in translation (as mentioned above), and replication [102,105,114]. Together they compose a *cis*-acting replication element (CRE) [102,105]. The CRE efficiently and specifically binds the human 40S ribosomal subunit, mainly at 5BSL3.1 and 5BSL3.3. This, along with the association of NS5B to 5BSL3.2, highlights the critical role of these domains in controlling the switch from translation to replication [113]. The balance between different functionalities performed by these three domains is achieved by preserving the proper conformational equilibrium. 5BSL3.1, 5BSL3.2, and 5BSL3.3 fold into a high-order, cruciform structure with dynamic properties [102], that promote the switch between different metastable structural states in the RNA molecule by virtue of different external stimuli. This renders domains 5BSL3.1, 5BSL3.2, and 5BSL3.3 organizers and regulators of replication initiation.

During translation, viral genome circularization mainly accounts for the establishment of protein bridges between both ends of the RNA. In the early steps of replication, the virus takes advantage of the different structural elements at either end of the RNA molecule to promote this circular isoform. Its acquisition is the initiation signal for viral HCV RNA synthesis (providing further proof of the versatility of viral RNA genomes). In HCV, indirect evidence suggests the existence of two long-range RNA-RNA contacts involving different functional domains that promote the formation of a closed-loop conformation (for a review, see [3]).

Our group provided the first evidence of HCV RNA circularization mediated by RNA-RNA contacts in 2009 [115]. We showed that the bulge of the 5BSL3.2 at the 3′ end of the ORF directly interacts with the apical loop of subdomain IIId within the IRES (Figure 4B). This interaction is stable in the absence of protein factors and induces conformational changes, not only in the directly interacting regions, but also in other distant elements of the viral genome, as demonstrated in replication competent RNA transcripts [86,116]. In subdomain IIId, the residues at the apical loop may appear partially occluded and non-accessible to the surrounding solvent because of the interaction with domain 5BSL3.2 [117]. This alternative structure may reduce the affinity of the IRES for the 40S subunit, promoting the escape of the ribosome from the IRES and impeding translation initiation. Structural tuning of subdomain IIId has also been reported by other authors [21]. Fricke et al. described that the stem-loop IIId swings and extends 5 nt downstream, displacing the position of the apical loop, which appears as an extension of the stem (Figure 3) [21]. The displacement of the trinucleotide GGG from the apical loop to the stem would lead to less efficiently translated viral genomes by impeding the proper interaction with 18S rRNA [67,118]. This alternative structure could coexist with the previously validated conformation. Additional efforts are required to discover the triggering stimuli that promote the acquisition of each.

The contact IIId-5BSL3.2 also induces a rearrangement event in the 3′X tail, favoring the two stem-loop conformation (Figure 2, Figure 4B and Figure 5) [116]. Such a structure is associated with reduced translation efficiency [100]. This supports the model in which the interaction between subdomain IIId and the 5BSL3.2 domain interferes with efficient viral protein synthesis, thus, favoring the initiation of the replication step [114]. The model is confirmed by the observation that nucleotides located in the bulge of the 5BSL3.2 domain specifically and efficiently interfere with HCV IRES-dependent translation [114]. This strongly contrasts with the hypothesis proposing that 5BSL3.2 is required for viral protein synthesis (see above). It is noteworthy that the mechanism by which the 5BSL3.2 domain switches from translation enhancer to translation inhibitor depends on long-range RNA-RNA interactions, either with the 3′X tail or the subdomain IIId. These contacts are established in an independent manner and may be promoted by different factors [104,106,119] that induce conformational rearrangements in the involved regions.

HCV RNA circularization during viral replication is also stabilized by the long-range interaction established between the apical loop of domain II in the IRES, and the DLS motif in the 3′X region (Figure 2 and Figure 4B) [21]. In the theoretical model proposed, the initial interaction would span 62 base pairs, although this would require the (unlikely) complete unfolding of domains I and II of the IRES, together with the destabilization of the 3′X tail. Recent studies have suggested the participation of residues 95-110, which overlap with the bulge of subdomain IIb in the IRES, in a long-range RNA-RNA interaction with positions 8528-8543 in the NS5B coding sequence region (which is rich in stem-loop domains) (Figure 2 and Figure 4B) [120]. Interestingly, the flexible bulge of subdomain IIb takes part in the translation-to-replication switch [121], but in addition acts as moderate replication repressor [122], suggesting a new regulatory function for this long-range interaction. Both interacting motifs, 95-110 and 8528-8543, are structurally inaccessible; therefore, the destabilisation of both stem-loops is required for an efficient interaction to occur. Since the interaction IIId-5BSL3.2 induces fine-structural tuning to different regions of the HCV genome [86,116], it seems likely that it should operate as an initial contact to promote further conformational rearrangements leading to the stabilization of other long-range interactions required to achieve the circularization of the viral genome needed for replication control. These observations, along with the fact that the interaction IIId-5BSL3.2 is a negative regulator of HCV translation, suggest that viral genome circularization is a complex and coordinated process mediated by long-range RNA-RNA interactions and stabilized by the participation of different cellular and viral factors. Together, these create the proper environment for the initiation and regulation of HCV RNA synthesis.

## 6. Viral RNA Synthesis Is Controlled by High-Order Structures of the Genomic RNA

The control of HCV replication was associated with the 3′UTR for a long time. However, it is now known that the 5′UTR and the coding region contain important elements for viral RNA production (Figure 2 and Figure 4C).

In addition to the evident role of the 5′UTR in viral translation, it promotes a significant enhancement of replication efficiency. Domains I and II at the very 5′ terminus of the HCV genome (Figure 3) are indispensable for RNA synthesis [123,124]. However, it is uncertain whether this activity relays on these domains of the positive or the negative strand, or both. The molecular mechanism by which the 5′UTR participates in RNA synthesis is also unknown. The specific recruitment of different protein and RNA factors [125], as well as the structural preservation of different signals involved in replication, are tempting hypotheses [118].

In the core coding region, the preservation of the conserved stem-loop SL588 is essential for efficient replication, pointing to this domain as a *cis*-acting replication domain (Figure 2 and Figure 4C) [22]. Although the mechanism by which this element works has not been experimentally validated, its involvement in a long-range interaction with the apical loop of the conserved SL427 has been proposed (Figure 4C). This again highlights the importance of RNA-RNA interactions in the control of the HCV cycle.

HCV replication requires the binding of the NS5B protein to the 5BSL3.2 domain, a process favored by the locally high concentration of the NS5B protein and the RNA genome [110,126]. The question remains as to how the 5BSL3.2 domain moves to recruit NS5B from interacting with subdomain IIId in the circular isoform. Again, long-range RNA-RNA contacts might provide an efficient mechanism. A highly conserved sequence motif named Alt was identified upstream of the 5BSL3.2 domain (Figure 4B) [127]. This sequence locates to the base of the stem-loop named 9110 and can base pair with the bulge of 5BSL3.2 (Figure 4C) [106,127]. The interaction promotes a conformational rearrangement at the 3′X tail toward the two stem-loop isoform, leaving a 3 nt overhang at the 3′ end of 3′SL1, which emerges via a slight displacement to generate the so-called 3′SL1′ (Figure 5) [99,107]. This folding state provides an appropriate environment for virus replication [128]. It should be noted that this Alt-5BSL3.2 contact overlaps with that involving subdomain IIId. Both interactions show constant dissociation values in the same range, and seem equally likely to occur [106]. Further, they are assumed to show an additive and enhancing effect on RNA replication by promoting a repressed translational state in the viral RNA (interaction IIId-5BSL3.2) and subsequently favoring the recruitment of the NS5B protein (interaction Alt-5BSL3.2). It seems likely that transitions between both contacts are influenced by additional, non-reported structural constraints and the presence of different cofactors. In this context, the specific binding of viral and cellular proteins to the 5BSL3.2 domain is well reported and may underlie the intricate mechanism that regulates viral RNA production [129,130,131,132,133].

The NS4B coding region also contains tools for the control of RNA replication (see Figure 2 and Figure 4C) [22]. For example, a functional domain named SL6038 has been described which can adopt two conformations: a cloverleaf-like structure that activates viral RNA synthesis, and a stem-loop isoform that interferes with the replication [22]. Similarly, stem-loop SL8001 in the NS5B coding region has been proposed to be a replication control device via its mediating conformational tuning events.

Besides its involvement in intramolecular interactions, the HCV RNA genome can take part in intermolecular contacts via the formation of genomic homodimeric particles with structurally unusual features, and in the absence of any protein factors (Figure 4C and Figure 5) [98]. Dimer formation in HCV is dependent on the palindromic DLS motif (Figure 5), which operates as a starting signal for the formation of a kissing complex involving the nucleotides exposed in the loops [98]. The initial complex can evolve, under certain conditions, towards an extended duplex conformation involving the whole DLS motif [119]. As a consequence, the 3′SL1 domain refolds to 3′SL1′, creating the optimal structural environment for NS5B activity [134].

Distant genomic regions influence HCV dimer formation efficiency. Dimer formation is controlled by the CRE at the 3′ end of the ORF (i.e., 5BSL3.1, 5BSL3.2, and 5BSL3.3), which improves the dimerization yield, and by the IRES at the 5′ end, which exerts an inhibitory effect on dimerization even in the presence of enhancer elements [135]. The high conservation rate of both the IRES and the CRE, and their essential roles in translation and replication, suggest that dimerization is critical for virus propagation. These observations confirm the strong influence of the CRE region (particularly 5BSL3.2) in the regulation of different steps of the viral cycle via its participation in the establishment of a complex network of contacts. Moreover, the latter work supported previously collected data suggesting that undiscovered IRES-3′X contacts [21,116] help in the acquisition of the closed-loop conformation during the replication step. In fact, this conformation would be preserved even in the negative strand since these interactions are predicted to take place as well, pointing to genome circularization mediated by distant RNA-RNA contacts as the core molecular mechanism in the regulation of the viral cycle, including the switching between events.

## 7. RNA Elements also Control Viral RNA Packaging

Virion assembly in HCV is a complex process that relies strongly on the interaction of the RNA genome with the capsid - or core - protein. It also requires the coordinated action of viral proteins and host factors (for a review, see [136]). To date, all the non-structural viral proteins have been shown to contribute to this process, though the mechanism by which virion assembly occurs remains elusive. The development of new virus culture systems in recent years has provided new evidence on the mechanism involved in the incorporation of viral genomes into new virions, but the full picture remains unclear.

The HCV assembly process appears to be spatially associated with lipid droplets, to which the core protein is linked [137]. Interaction with the RNA genome requires a structural signal, located at the 3′ end of the viral genome, which is recognized by specific domains of the core protein (Figure 2 and Figure 4D). This mechanism ensures the incorporation of full-length genomes only [138]; no packaging of replication-defective HCV genomes can take place. The maintenance of the structural signal at the 3′ terminus, either via viral NS3 helicase [139] or by preserving the proper balance between different RNA-RNA interactions [3,26], is therefore, indispensable for correct encapsidation. In this context, it is noteworthy that the 5BSL3.2 domain diminishes the affinity of the 3′UTR for the core protein, confirming previous observations that point to a conformational rearrangement at the 3′ end of the viral genome mediated by the interaction 5BSL3.2-3′X [99,100,104,107,116,135]. Hence, the preservation of the structure at the 3′X tail for successful viral packaging is likely to prevent its interaction with the 5BSL3.2 domain. Again, the HCV genome takes advantage of its complex folding to create dynamic interaction networks that can displace the equilibrium towards the desired step of the infective cycle. For example, the element SL8670 [20], which is involved in virus production, but not in RNA synthesis, is predicted to interact with a part of the DLS motif [26], suggesting a control mechanism for RNA packaging (Figure 2 and Figure 4D).

The 3′X structural signal is not the only actor required for efficient HCV assembly. In contrast to the many RNA viruses that use large, stable RNA structural elements with high-affinity for the core protein as the encapsidation signal, HCV seems to takes advantage of multiple domains with low affinity for the core protein as a means of presenting the intact, full-length genome to the assembly complex (Figure 4D) [140]. Such domains are dispersed throughout the genome to act in a cooperative manner. Using a SELEX (systematic evolution of ligands by exponential enrichment) methodology, Stewart et al. [141] demonstrated the existence of different conserved stem-loops located in the core and the non-structural protein-coding sequence that is involved in the packaging (see Figure 2 and Figure 4D). Other authors have also noted different stem-loops in the core and NS5B coding sequences to be important in virion production [20,22]. The results of the above SELEX analysis also confirmed the absence of encapsidation signals in the E1 and E2 coding regions [142,143]. All of the identified stem-loops in combination, but not independently, are required for virion production, though they are not essential for HCV translation and replication, suggesting them to be specific, cooperative signals for genome encapsidation. The preservation of folding at precise locations is, therefore, also important for RNA genome packaging and virion budding [20,22,141].

## 8. Genome Circularization Strategies in RNA Viruses Mediated by RNA-RNA Contacts

Viral genome circularization is not exclusive to hepaciviruses. Indeed, circular RNAs are represented in all kingdoms of life. In RNA viruses, however, the ubiquitous circularization process acquires new importance, acting as a regulatory tool in *cis*. It is especially important in viruses lacking the cap structure (e.g., picornaviruses), the polyA tail (e.g., reoviruses) or both (e.g., hepacivirus). In good agreement with the HCV model, the 5′-3′ end communications in other RNA viruses involve elements critical for translation and replication, making the closed-loop topology indispensable for viral fitness. Since translation and replication are mutually exclusive, it is also likely that circular isoforms operate as elements that manage the transitions between different stages of the viral cycle.

In some cases, such as in polioviruses and reoviruses, RNA genome circularization occurs via a protein-protein bridge [144,145]. In flaviviruses (e.g., Dengue virus [DENV]) and retroviruses (e.g., human immunodeficiency virus [HIV]), however, complementary sequences located at the 5′ and the 3′ ends of the viral genome can base pair to establish a long-range contact. This initial contact can be further stabilized by the recruitment of viral and/or cellular proteins, but more importantly, via additional RNA-RNA interactions [12,146,147].

Retroviral replication involves a reverse transcription step with two obligatory DNA strand transfers; these are needed to generate the complete full-length RNA genome, which is then packaged. To facilitate these strand transfer events, the virus uses RNA circularization dependent on RNA-RNA interactions [12,148]. In HIV-1, one of the most studied models, several interactions have been proposed to participate in the acquisition of a closed-loop conformation (Figure 6A). One involves a palindromic sequence motif in the TAR element. The TAR hairpin is a highly conserved domain with an essential role in viral replication. Since it is located at both ends of the viral genome, it is thought to mediate direct circularization. The other interaction connects sequence motifs in the *gag* gene at the 5′ end of the open reading frame with the 3′ terminal U3R region [12,148]. In both cases, it is important to note that the elements potentially involved in the circularization of the genome are evolutionarily conserved across different HIV-1 subtypes, suggesting that the circular isoform is actively used.

In flaviviruses, end-to-end genome connections rely on at least three specific interactions involving complementary sequence motifs located at essential domains at both ends of the viral RNA (Figure 6B). The pairs of nucleotide motifs include the so-called CYC element, the UAR pair, and the DAR sequence motifs (for a review, see [10]). These pairs are established in a sequential manner, allowing for cooperative and efficient formation of the closed loop, which is then stabilized by the recruitment of different host factors to improve the final yield [10]. Though low conservation rates have been reported for some of the sequences involved, the cyclization process is a universal and essential event among flaviviruses [149,150].

This brief overview reflects the universality of the circularization process in RNA viruses and its value in controlling the viral cycle. It is noteworthy that circular isoforms emerge under certain stimuli and may coexist with the linear form. Preserving the balance between different conformations helps ensure viral fitness and propagation.

## 9. Conclusions and Perspectives

Protein coding-information represents only a small portion of the genetic information carried in the genome. Viral RNA genomes use an information coding system involving structural genomic domains that overlap the protein-coding system. These domains express their information by establishing a dynamic network of long-distance RNA-RNA interactions - the interactome - and the recruitment of cellular and viral factors. It is this network that governs the execution of the viral cycle. In the case of the HCV RNA genome, it seems clear that the 5BSL3.2 structural domain within the CRE, located at the 3′ end of the ORF, plays a central role in the management of the RNA-RNA interactome, and therefore, in the fine regulation of the viral cycle. During the early phase of infection, the RNA genome mainly functions as mRNA, and is therefore, dedicated to translation. However, the recruitment of viral RNA polymerase by the structural domains at the 3′ end of the genome, and the consequent modification of RNA-RNA interactions, leads 5BSL3.2 to repress IRES translation initiation. The 5BSL3.2 domain also promotes replication, increasing the number of viral genomes and favoring the formation of genomic RNA dimers. These replication competent-conformers remain, however, incompetent for packaging. The dissociation of the dimers yields competent forms for translation and consequent packaging.

Exploring the interactome of other related RNA viruses and their functionality should provide further information on the regulation of the viral cycle and provide a means of identifying potential targets for the development of antiviral strategies. It should be anticipated that the lack of structural RNA domain conservation among different viruses means that different RNA viruses, even when closely related, will use different strategies involving different RNA domains to achieve the goal of regulating the essential steps of their infective cycle.

## Figures and Tables

**Figure 1 ijms-21-01479-f001:**
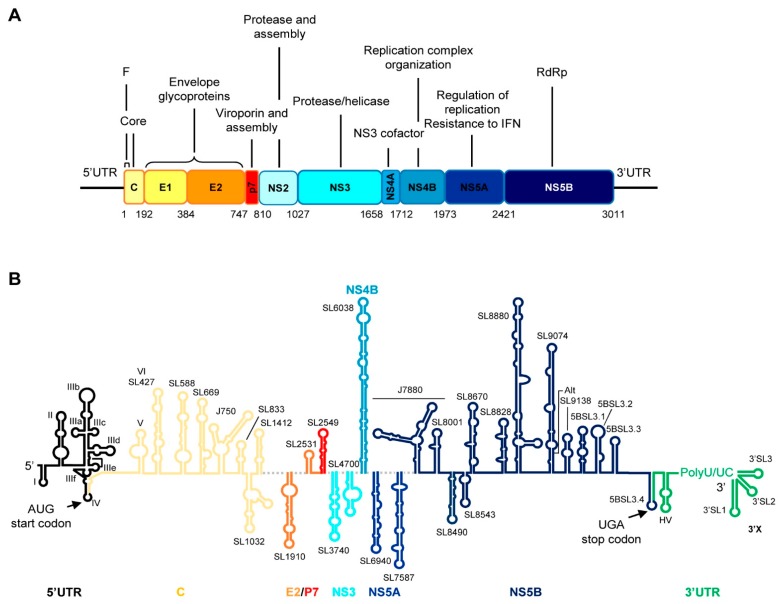
Genetic organization and structurally conserved RNA domains of the hepatitis C virus (HCV) genome. (**A**) Diagram showing the genetic organization of the viral genome with the 5′ and 3′ UTRs are flanking the single open reading frame (ORF). Viral structural and non-structural (NS) proteins and their functions are indicated. The numbering corresponds to codon positions in the ORF according to the HCV Con1 isolate, genotype 1b. (**B**) Diagram of the secondary structure model of the viral RNA genome. Known conserved secondary structural domains are designated by their names. The color code and labels at the bottom indicate where each stem-loop is located. The translation starts and stop codons are marked by arrows.

**Figure 2 ijms-21-01479-f002:**
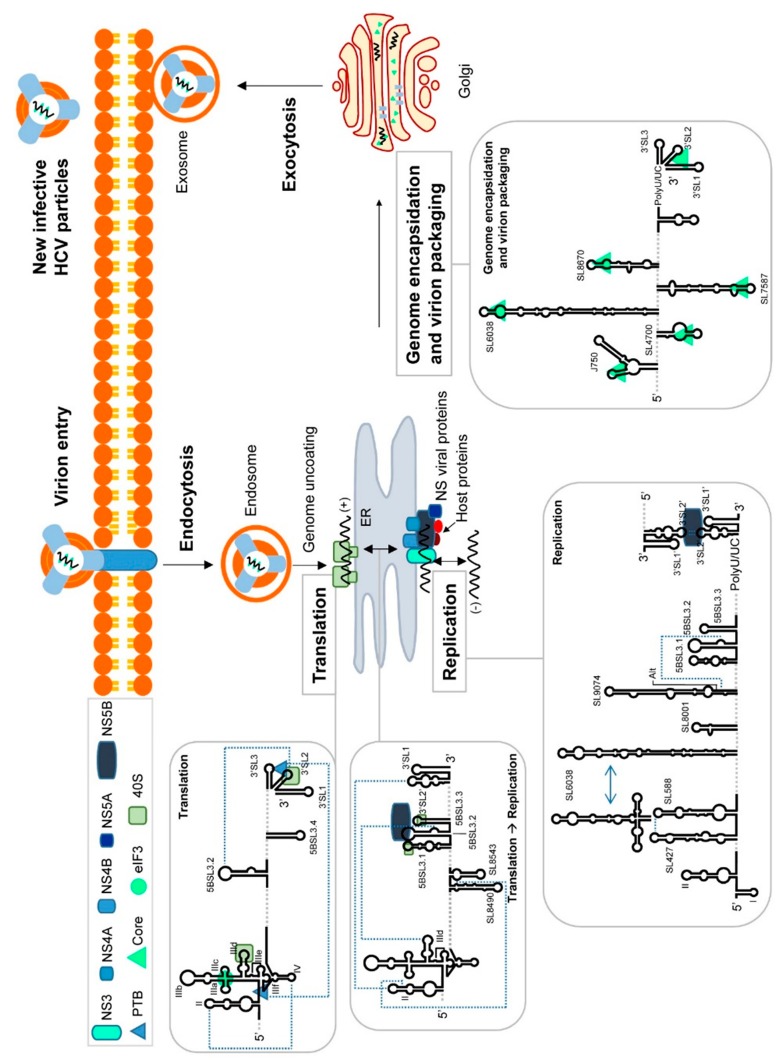
Long-range RNA-RNA interactions and RNA structural domains control the HCV intracellular cycle. The figure shows a model of the viral cycle, outlining the participation of functional genomic domains and the RNA-RNA interactome in every step. Briefly, following the recognition of the viral envelope proteins by receptors on the hepatocyte surface, entry occurs by clathrin-mediated endocytosis. The endosome membrane and the viral envelope fuse and the capsid become disorganized (uncoating) - a process that requires the low pH of the endosome interior. The genomic RNA is directly translated on the surface of the endoplasmic reticulum (ER) in an IRES-dependent manner. IRES subdomain IIId is occupied by the 40S ribosomal subunit, impeding the interaction IIId-5BSL3.2, but favoring the contact 5BSL3.2-3′SL2. The interaction established between IRES domains II and IV provides an optimal structural environment for the correct positioning of the translation start codon. The translation is enhanced by the acquisition of a circular isoform mediated by the oligomerization of PTB which binds to the IRES and the 3′X tail. The synthesized polyprotein is co- and post-translationally processed to generate all the viral factors required for replication. Switching from translation to replication requires the formation of a closed-loop conformation mainly dependent on long-range RNA-RNA interactions: Domain II-SL8523; domain II-3′SL2′ and subdomain IIId-5BSL3.2. These contacts promote a translationally repressed-state and enhance replication via the recruitment of the NS5B viral polymerase and the 40S subunit to the functional region at the 3′ of the ORF. Viral RNA synthesis occurs in the replication complexes on the surface of the endoplasmic reticulum; this requires multiple RNA domains, but also important conformational rearrangements at the 3′X promoted by the interaction 5BSL3.2-Alt. This contact induces a structure at the 3′ end of the viral genome that is susceptible to forming genomic dimers, an optimum substrate for the NS5B protein. Finally, encapsidation requires multiple stem-loops that work in a cooperative manner to ensure the packaging of full-length, intact viral RNA genomes. This process occurs in the ER and in the Golgi apparatus, yielding mature virions that are released to the extracellular medium by conventional exocytosis.

**Figure 3 ijms-21-01479-f003:**
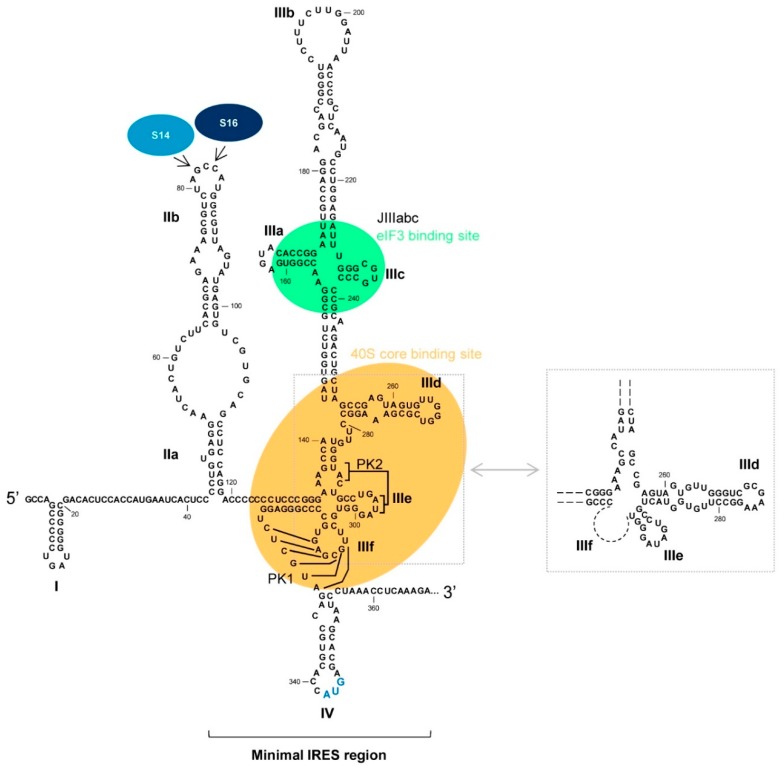
The HCV IRES element. Sequence and secondary structure of the 5′UTR in the HCV genome, including the minimal internal ribosome entry site (IRES). Domains involved in the interaction with eIF3 and the 40S ribosomal subunit are marked in green and orange respectively. Potential interaction sites of ribosomal proteins S14 and S16, depicted as blue ovals, are indicated with arrows. The translation start codon is revealed in enlarged blue lettering. An alternative folding state of subdomain IIId is shown, mediated by the slight displacement of five nucleotides that promotes a significant rearrangement at the base of domain III. PK, pseudoknot. Numbering corresponds to nucleotide positions of HCV Con1 isolate, genotype 1b.

**Figure 4 ijms-21-01479-f004:**
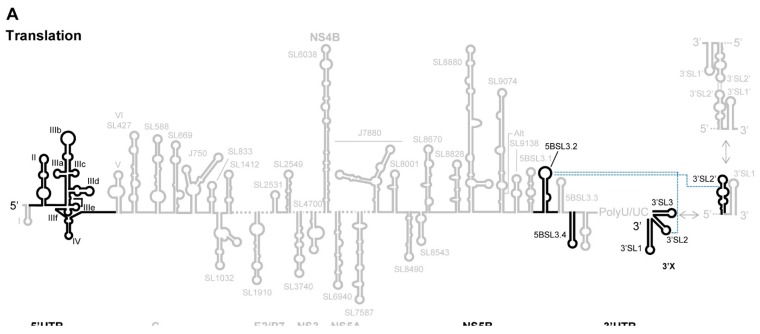
(**A**). RNA-RNA interactome in the HCV genome. RNA-RNA interactome in the HCV genome. Secondary structure model of the RNA genome and the structural tuning events required for the execution of the HCV cycle. Structural domains are designated by their names. Diagrams show the RNA domains, indicated with thick black lines, and their interactions, indicated by dashed blue lines. Their roles for different steps of the viral cycle – translation (**A**), replication (**C**) and transitions between them (**B**), as well as RNA genome packaging (**D**) – have been proposed.

**Figure 5 ijms-21-01479-f005:**
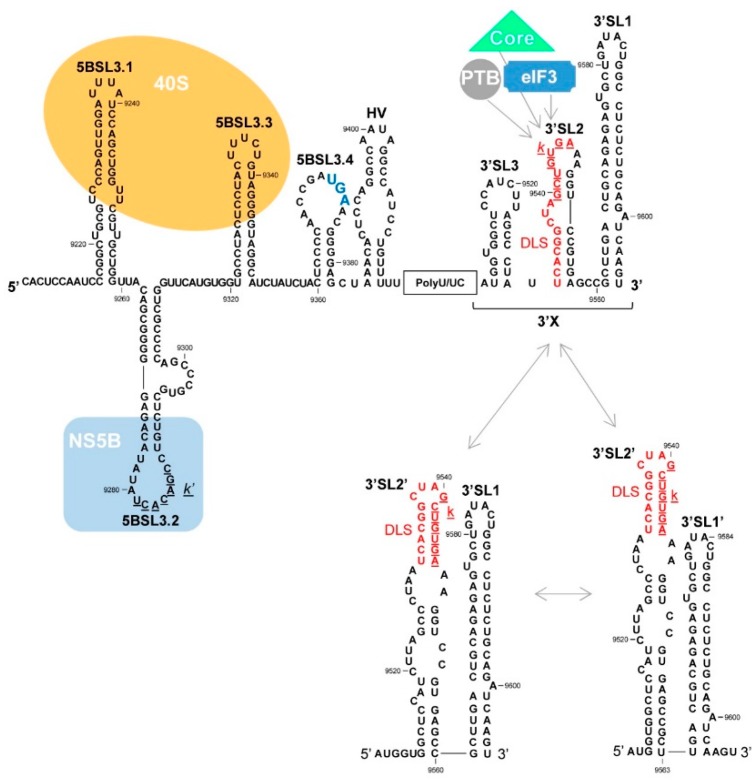
The 3′ end of the HCV genome. This figure shows the sequence and the widely accepted secondary structure model of the genomic 3′UTR and the upstream functional region containing the 5BSL3.1, 5BSL3.2, 5BSL3.3 and 5BSL3.4 domains. The theoretical alternative conformations acquired by the 3′X tail are also shown. The palindromic motif involved in HCV genome dimerization (DLS, dimer linkage sequence) is shown in red. The k and k’ sequences in the DLS and apical loop of the 5BSL3.2 domain respectively are underlined; these are required by both domains for their interaction activity. The translation stop codon is shown by enlarged blue lettering. The binding sites for viral and cellular proteins are indicated by colored backgrounds. Nucleotide numbering is as in Figure 3.

**Figure 6 ijms-21-01479-f006:**
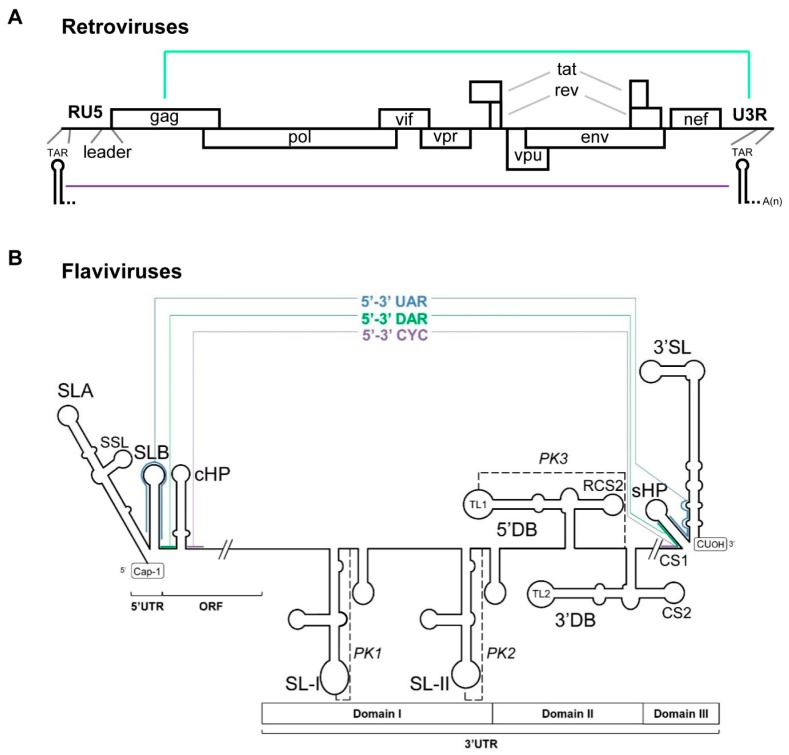
Long-range RNA–RNA contacts in representative RNA viruses with linear genomes. This figure shows the proposed conserved secondary structural elements and sequence motifs within the 5′ and 3′ ends of retroviruses, such as HIV-1 (**A**) and DENV-2 (representative of flaviviruses) (**B**) involved in genome circularization. The ORF and the UTRs (RU5 and U3R for retroviruses) are indicated. Thin colored lines denote long-distance RNA–RNA interactions between genomic termini. The pseudoknot elements (PK1, PK2, and PK3) are indicated by dashed black lines.

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
