# Peer review of "The Role of the RNA-RNA Interactome in the Hepatitis C Virus Life Cycle"

_ijms, 2020, doi:10.3390/ijms21041479_

Round 1

Reviewer 1 Report

The paper by Romero-López and Berzal-Herranz is a comprehensive review on HCV genome structure and RNA-RNA interactions important for various aspects of the biology of the virus. The paper is very detailed end is important because HCV still affects a significant number of populations world-wide. The uathors discus potential drug targets and current drugs for HCV are expensive and do not reach needy individuals in resource-poor countries. Are there RNA structures that could be exploited for vaccine development? The authors also discus a riboswitch phenomenon, which is very interesting. What could be the ligands or external stimuli that could be involved in modulating this switch?

Author Response

We acknowledge the referee for the constructive comments, and for considering that our work can contribute to a better understanding of the HCV biology.

  1. Are there RNA structures that could be exploited for vaccine development?

This is a very interesting view of the theme. Although antibodies have been traditionally targeted against proteins, different RNA structural motifs have been proved to act as efficient antigens (Jung et al., 2014, RNA, 20(6): 805–814). However, the development of successful vaccines targeting these structural motifs is still an ideal, mainly due to the fact that viral RNA remains encapsidated –hence it is not accessible to solvent- during the extracellular stages of the infection. Nevertheless, antibodies against structural elements of RNA genomes could be useful diagnostic tools for genotype testing.

  1. The authors also discuss a riboswitch phenomenon, which is very interesting. What could be the ligands or external stimuli that could be involved in modulating this switch?

It is a fascinating field of study, which unfortunately, remains to be solved. To our knowledge, the specific external stimuli required for the reported HCV switching mechanisms remain elusive. Therefore, we cannot make any proposal in the manuscript at this respect.

Reviewer 2 Report

Dear authors,

congratulations for your review. HCV genome was described with specific molecular data allowing us to understand more about HCV viral cycle and RNA-RNA interactions. I only have two suggestions to improve manuscript introduction:

  1. Lines 56-58: "Treatment with two new viral protease inhibitors in combination with pegylated interferon-α and ribavirin now allow the infection to be controlled, significantly improving patient prognosis". You should expose what DAAs are used in clinical routine. Interferon-free therapies with NS5A and NS5B DAAs have been incorporated in clinical protocols?
  2. Lines 77-78: "Like most RNA viruses, HCV is highly variable from a genetic point of view, with seven different 78 genotypes showing more than 30% nucleotide sequence divergence among them [12]". According to Borgia SM et al. (2018), a novel HCV genotype 8 was identified. Reference: https://www.ncbi.nlm.nih.gov/pubmed/29982508

Author Response

We thank the reviewer for the helpful comments, which have improved the manuscript.

  1. Lines 56-58: "Treatment with two new viral protease inhibitors in combination with pegylated interferon-α and ribavirin now allow the infection to be controlled, significantly improving patient prognosis". You should expose what DAAs are used in clinical routine.

We have modified the text according to reviewer’s suggestion.

Interferon-free therapies with NS5A and NS5B DAAs have been incorporated in clinical protocols?

Yes, they have. In fact, treatments in chronic patients avoid the use of pegylated-interferon in order to minimize long-term, hard adverse effects (Alazard-Dany et al., 2019, Viruses, 11(1): 30).

  1. Lines 77-78: "Like most RNA viruses, HCV is highly variable from a genetic point of view, with seven different 78 genotypes showing more than 30% nucleotide sequence divergence among them [12]". According to Borgia SM et al. (2018), a novel HCV genotype 8 was identified. Reference: https://www.ncbi.nlm.nih.gov/pubmed/29982508.

We thank the reviewer for this point. The reference has been included in the manuscript and the information provided has been modified accordingly.